# Diatom abundance in the polar oceans is predicted by genome size

**Wade R. Roberts**  *, Adam M. Siepielski, Andrew J. Alverson*

Department of Biological Sciences, University of Arkansas, Fayetteville, Arkansas, United States of America

* wader@uark.edu (WRR); aja@uark.edu (AJA)

**Data Availability Statement:** Sequencing reads and genome assemblies are available from NCBI BioProject PRJNA825288. Datasets and code are available from Zenodo (DOI:10.5281/zenodo.

## Abstract

A principal goal in ecology is to identify the determinants of species abundances in nature. Body size has emerged as a fundamental and repeatable predictor of abundance, with smaller organisms occurring in greater numbers than larger ones. A biogeographic component, known as Bergmann's rule, describes the preponderance, across taxonomic groups, of larger-bodied organisms in colder areas. Although undeniably important, the extent to which body size is the key trait underlying these patterns is unclear. We explored these questions in diatoms, unicellular algae of global importance for their roles in carbon fixation and energy flow through marine food webs. Using a phylogenomic dataset from a single lineage with worldwide distribution, we found that body size (cell volume) was strongly correlated with genome size, which varied by 50-fold across species and was driven by differences in the amount of repetitive DNA. However, directional models identified temperature and genome size, not cell size, as having the greatest influence on maximum population growth rate. A global metabarcoding dataset further identified genome size as a strong predictor of species abundance in the ocean, but only in colder regions at high and low latitudes where diatoms with large genomes dominated, a pattern consistent with Bergmann's rule. Although species abundances are shaped by myriad interacting abiotic and biotic factors, genome size alone was a remarkably strong predictor of abundance. Taken together, these results highlight the cascading cellular and ecological consequences of macroevolutionary changes in an emergent trait, genome size, one of the most fundamental and irreducible properties of an organism.

## Introduction

The abundance of species in nature is a central feature of all life. Because of this centrality, a principal goal of ecology is to understand what determines organismal abundance [1–3]. Theoretical studies have developed an extensive body of work to understand how demographic parameters (e.g., birth and death rates) affect species abundances [4–6], while observational and experimental studies have identified key abiotic (e.g., nutrient supply) and biotic factors (e.g., species interactions such as competition and predation) that shape the abundances of organisms from local to global scales [7–11]. Another equally large body of literature has sought to identify the key intrinsic features of organisms that shape their abundance [12–14].

12608914). Accession numbers for additional public datasets used in this study are available in S1 Table.

**Funding:** This work was supported by the National Science Foundation (DEB 1651087 to A.J.A.). The funders had no role in the study design, data collection and analysis, decision to publish, or preparation of the manuscript.

**Competing interests:** The authors have declared that no competing interests exist.

**Abbreviations:** EPA, evolutionary placement algorithm; ESS, effective sample size; NCMA, National Center for Marine Algae and Microbiota; OTU, operational taxonomic unit; PGLS, phylogenetic generalized least square; RCC, Roscoff Culture Collection.

Among these efforts, the size of an organism has emerged as a fundamental and repeatable predictor of abundance—smaller organisms occur in greater numbers than larger ones [15]. This relationship occurs across unicellular and multicellular lineages, and in terrestrial and aquatic ecosystems [15–17]. A biogeographic component, known as Bergmann's rule, describes an association between body size and temperature, wherein larger-bodied organisms are found in colder environments and smaller organisms in warmer ones [18,19]. Thus, body size and temperature are frequently woven together as key explanations for organismal abundance. The repeatability of these associations, which link a fundamental organismal trait to its abundance and thermal environment, are heralded as a widespread feature of life on Earth [20,21]. But key questions remain, such as what determines size and whether size alone is the most basic intrinsic, ecologically determinant feature of an organism. For multicellular species, size is a complex trait confounded by tissue differentiation, life history, and development [22–24]. For unicellular organisms, which constitute the bulk of life on Earth, their size may be fundamentally shaped by a single intrinsic feature, the size of their genome [23,25]. Across eukaryotes, genome size varies by many orders of magnitude and is correlated with numerous traits of ecological importance, including body size, metabolism, and life history [16,26,27]. As a result, genome size may have important cascading effects on organismal abundance and, ultimately, ecosystem function [28–30].

To test this hypothesis, we asked whether genome size can predict patterns of diatom abundance across the world's oceans. Diatoms are single-celled primary producers that account for 20% of global primary production and are keystone species in marine food webs [31]. We traced the history of genome evolution in one of the most diverse and abundant lineages of marine planktonic diatoms, Thalassiosirales [32,33], to characterize the determinants of genome size on evolutionary timescales. Although a simple association between genome size and body size (cell volume) seems intuitive, a longstanding question is whether genome size drives cell volume, or whether cell volume—an ecologically important and putatively adaptive trait—drives changes in genome size [22,24,34]. We used phylogenetic path analysis to test competing directional hypotheses about the relationship between these 2 traits, which have the potential to shape key population demographic parameters that should, in turn, shape species abundances in accord with basic population ecology theory [4–6]. We then used a large meta-barcoding database to determine whether genome size predicts geographic patterns of diatom abundance and temperature associations in the global ocean. Our results identified genome size as a strong predictor of global patterns of phytoplankton species abundance. Thus, in the absence of any additional information, this single, emergent property of an organism can help us understand species abundance in the wild.

## Results

### Repetitive DNA underlies broad variation in genome size

We characterized the genomes of 67 newly ($n = 46$) and previously sequenced ($n = 21$) diatom strains, representing 51 species of Thalassiosirales (S1 Table). Haploid genome size varied by nearly 50-fold, from 33 Mb in *Cyclotella nana* to 1.5 Gb in *Thalassiosira tumida* (Fig 1) and showed strong phylogenetic signal (Pagel's $\lambda = 0.998$, $P < 0.001$). Estimates of haploid genome size based on $k$-mer counting and sequencing coverage were similar and strongly correlated (Spearman's $\rho = 0.984$, $P < 0.001$) (S1 Fig). Our estimates of genome size were similar for the 3 strains in our dataset with genome size estimates from flow cytometry (S1 Table). For example, our estimate for *Cyclotella nana* CCMP1335 was 33 Mb, while flow cytometry estimated it at 36 Mb (S1 Table). The $k$-mer-based method was unable to estimate genome sizes for 5 taxa, so our results are based on the coverage-based dataset unless stated otherwise. Thalassiosirales

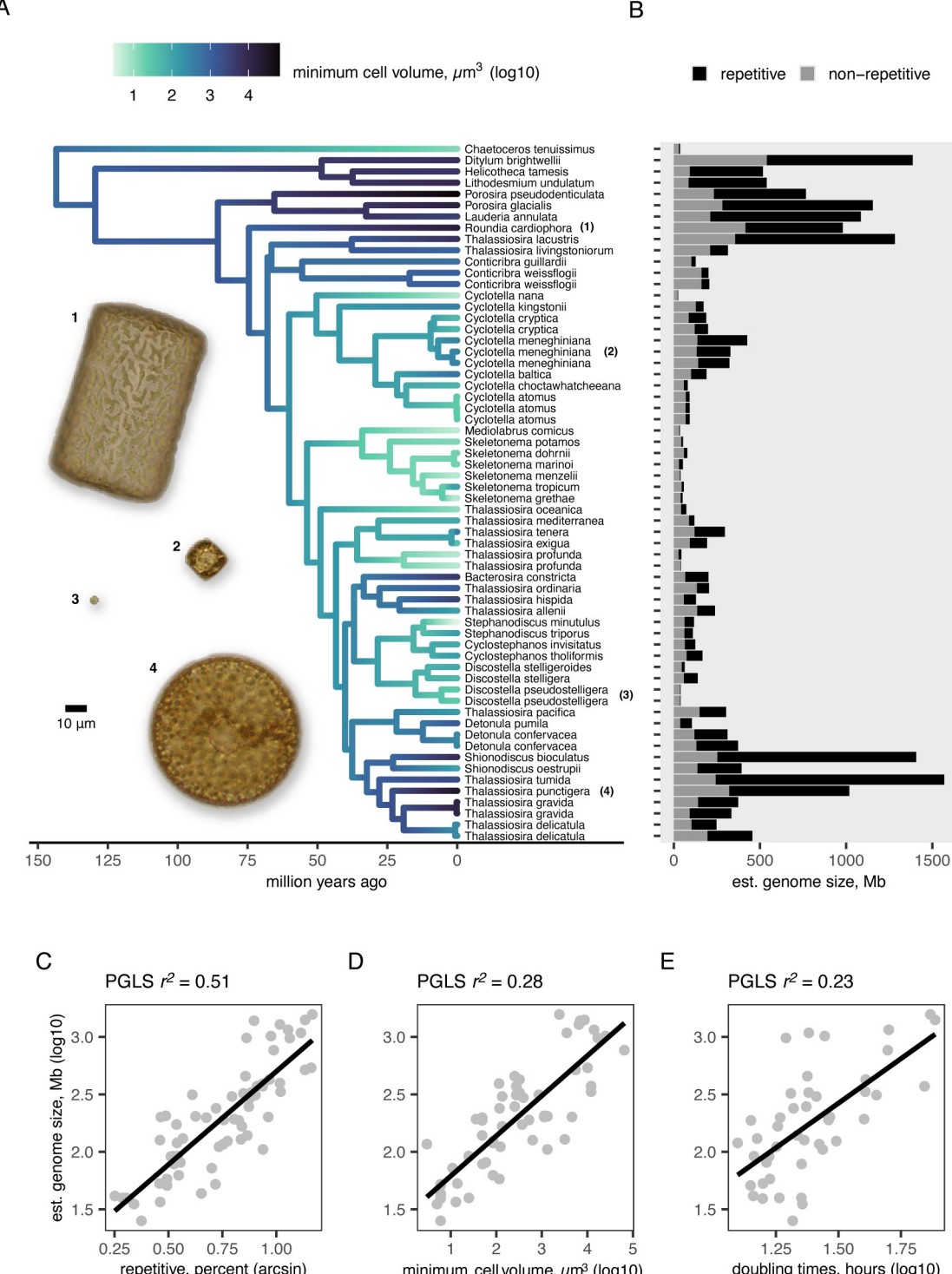

**Fig 1. Cell volume and genome size vary widely across diatoms.** **(A)** A time-calibrated phylogeny of the diatom order Thalassiosirales, modified from [35] and with branches colored by minimum cell volume. Light micrographs of live cells illustrate the broad variation in cell volume across the lineage. **(B)** Bar plots show the estimated genome size and proportions of non-repetitive and repetitive DNA in each genome. Panels C–E show PGLSs models predicting genome size with **(C)** percentage of repetitive DNA, **(D)** minimum cell volume, and **(E)** measured cell doubling time. Black lines show the estimated regression coefficients. The data and code needed to generate this figure can be found in https://doi.org/10.5281/zenodo.12608914.

includes marine and freshwater species [35], but there was no significant difference in genome size between diatoms from the 2 environments (Wilcoxon rank sum test, $P$ = 0.125) (S2 Fig).

Genome size was strongly correlated with repetitive DNA content (phylogenetic generalized least squares [PGLS] $r^2$ = 0.51, $P$ < 0.001) (Figs 1 and S3). The percentage of the genome composed of repetitive DNA ranged from 6% in *Thalassiosira profunda* (genome size: 41 Mb) to 85% in *Thalassiosira tumida* (genome size: 1.5 Gb) (S1 Table). Among the different classes of repetitive DNA, unclassified repetitive elements constituted the largest fraction of most genomes (S4 Fig). These are repetitive sequences that could not be classified into known repeat classes, likely due to the paucity of large diatom genomes that have been sequenced to date. The different classes of repetitive elements increased more-or-less proportionally in larger genomes, such that no single class of repetitive DNA disproportionately drove increases in genome size (S4 Fig). There was no association between haploid genome size and the average length of genes, exons, or introns (S3 Fig), nor the presence of polyploidy (S1 Table). Previous studies have linked GC content to genome size variation in both multicellular and unicellular organisms [36], but genome size was weakly negatively correlated with average genome-wide GC content in these diatoms (PGLS $r^2$ = 0.08, $P$ = 0.013) (S3 Fig).

## Genome size affects cell size and growth rate

Genome size is strongly correlated with body size, measured as cell volume, in microbial eukaryotes [23]. Although the extent to which increases in genome size require commensurate increases in nuclear and cell volumes is unclear [22], genome size should exert its greatest influence on the minimum volume of a cell. To test whether genome size predicts cell volume in diatoms, we compiled minimum and maximum volumes for the 51 species in our dataset. Maximum cell volume varied by 5 orders of magnitude across species and minimum cell volume varied by 4 (Figs 1 and S5 and S2 Table). Increased genome size was associated with increases in both minimum (PGLS $r^2$ = 0.28, $P$ < 0.001) and maximum cell volume (PGLS $r^2$ = 0.53, $P$ < 0.001) (Figs 1 and S5). We measured maximum growth rates for the species in our study (S1 Table) to test whether genome size is a predictor of cell division rate and found that species with larger genomes did indeed have longer doubling times (i.e., slower growth rates) (PGLS $r^2$ = 0.42, $P$ < 0.001) (S6 Fig). Temperature has profound effects on cellular metabolism and growth rate in both multicellular and unicellular organisms [37], and the addition of temperature to genome size as a predictor of growth rate led to substantial improvement in model fit (PGLS $r^2$ = 0.73, $P$ < 0.01) (S6 Fig). Here, lower temperatures and larger genomes were both associated with decreased growth rate (S6 Fig).

Across the tree of life, genome size and body size are strongly correlated with growth rate, nutrient usage, and other life history traits, but causal relationships and trade-offs among these and other correlated traits are not always clear [22,34,38,39]. Although causality cannot be inferred directly from comparative analyses of observational data, we can test the relative support for alternative models. To that end, we used phylogenetic path analysis—a type of structural equation modeling that allows for the evaluation of causal hypotheses from empirical data—to test competing hypotheses about the effects of 4 variables on growth rate: genome size, body size (minimum cell volume), temperature, and genomic GC content. We generated 14 alternative hypotheses (i.e., sets of directional relationships) to test whether genome size has no effect (null models), direct effects (direct models), or indirect effects (indirect models) on growth rate (cell doubling time) (S7 Fig). Using coverage-based genome size estimates, 3 models (direct1, direct2, and direct4 in S3 Table) were equally supported, with ΔCICc values <2 and $P$ values >0.2, indicating good fit to the data (S3 Table). In the best-fit model, direct4, genome size directly affects cell volume and doubling time, and temperature directly affects

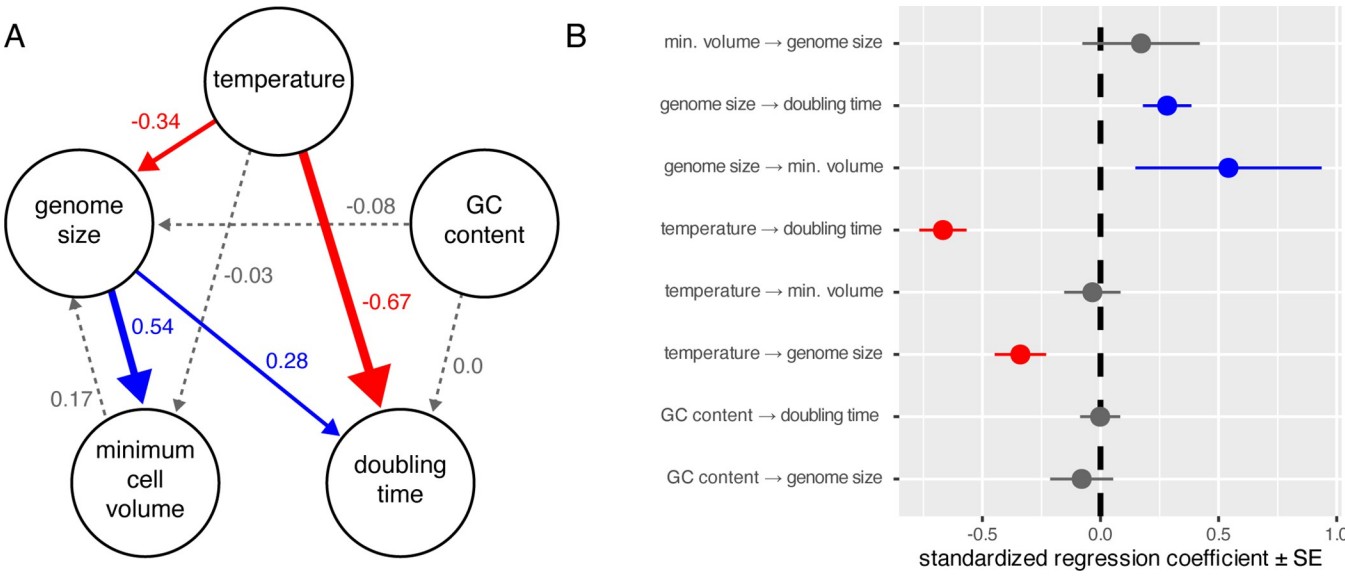

**Fig 2. Genome size affects cell volume and doubling time in diatoms. (A)** Average model from phylogenetic path analysis using coverage-based genome size estimates. Arrow color and width represent the direction and magnitude of regression coefficients, indicated by numeric labels (positive: blue; negative: red; nonsignificant: gray). Full lines show coefficients that differ significantly from 0, whereas dotted lines overlap with 0. **(B)** Standardized regression coefficients and their standard errors (SE) for paths in the model. The data and code needed to generate this figure can be found in https://doi.org/10.5281/zenodo. 12608914.

genome size and doubling time (S8 Fig). The direct1 and direct2 models remove the effect of GC content on genome size (S7 Fig). The main difference between the top 2 models (direct4 and direct2) is whether genome size impacts cell volume (direct4) or vice versa (direct2). Averaging the top 3 models resulted in a larger path coefficient for genome size affecting cell volume (0.54 versus 0.17) (Fig 2). Finally, testing the same 14 models with the *k*-mer instead of coverage-based genome size estimates gave 6 models (including direct4) with equally strong support (S3 Table). Importantly, all 6 models support genome size directly impacting cell volume, adding further support for the hypothesis that genome size influences cell volume, not the reverse (S7 and S8 Figs). The best-fit (indirect2) and average models using *k*-mer-based genome sizes both suggested that genome size affects doubling times but only indirectly via effects on cell volume, rather than the direct effect of genome size on doubling time supported by coverage-based genome size estimates (S8 Fig).

## Genome size, biogeography, and temperature impact diatom abundance in the ocean

Taking advantage of the global metabarcoding database from the *Tara* Oceans expedition [40], we built Bayesian models to test whether genome size influences relative species abundance in the ocean (Fig 3). Using 2 taxonomic assignment methods for operational taxonomic unit (OTU) sequences, we identified 28 species from our study that were also present in ≥10 samples of the *Tara* Oceans database. This allowed us to test whether latitude, ocean region, and/ or ocean temperature interact with genome size to affect species abundance. Latitude had a significant nonlinear interaction with genome size on species abundance (Fig 3C)—species with larger genomes were more abundant at high latitudes, and species with smaller genomes were more abundant at lower latitudes (Fig 3C). The 2 coldest ocean regions, the Arctic and Southern Oceans, were the only ones with a significant positive regression coefficient relating genome size to abundance, whereas all other regions had either no effect or a negative effect

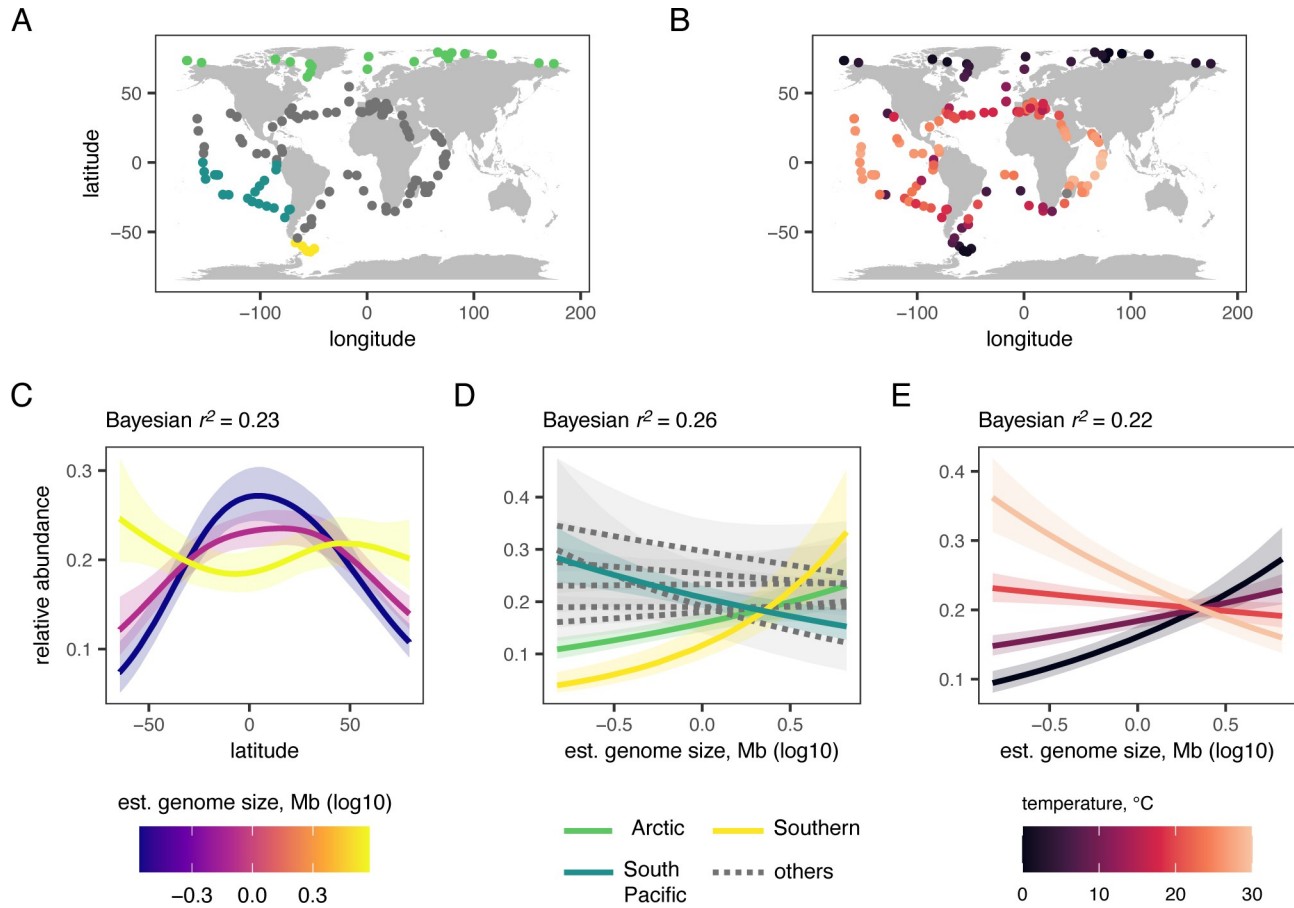

**Fig 3. Latitude, ocean region, and temperature interact to shape genome size—abundance relationships. (A, B)** Maps showing the locations of the 210 sampling stations from the *Tara* Oceans expedition, with points colored to highlight locations within the **(A)** Arctic, Southern, and South Pacific Oceans or **(B)** the temperature of each location at the time of collection. The base layer for the maps is from https://cran.r-project.org/web/packages/maps/index.html. Panels C–E show Bayesian multilevel regression models predicting relative species abundance by the interaction of genome size with **(C)** latitude, **(D)** ocean region, or **(E)** temperature. Nonlinear effects of latitude were modeled in **(C)** using a generalized additive model. Significant estimates for the Arctic, Southern, and South Pacific Oceans are shown with solid lines in **(D)**. Nonsignificant estimates for the other ocean regions are shown with dotted lines in **(D)**. Although all predictors are treated as continuous in **(E)**, we used the model to predict the interactive effect of 4 temperatures (0, 10, 20, and 30°C) with genome size on species abundance. The data and code to generate this figure can be found in https://doi.org/10.5281/zenodo.12608914.

(South Pacific Ocean) of genome size on species abundance (Fig 3D). We tested the effects of temperature directly and found a significant interaction with genome size to predict abundance, in which species with larger genomes were more abundant in colder temperatures (Fig 3E). These results were replicated with a smaller dataset (22 species) of OTUs classified using an alternate method of taxonomic assignment (S9 Fig).

## Discussion

One of the most basic and defining ecological properties of a species is its abundance in the environment, which is shaped by numerous interacting abiotic and biotic factors [2,6,9]. As a result, considerable attention has been paid to identifying key ecological processes that define a simple, sufficient, and generalizable ecological model explaining species abundance [11]. Rather than focusing on ecological processes, we sought instead to determine whether a fundamental intrinsic property of an organism—the size of its genome—can explain abundance and an associated vital rate, population growth [15,41].

Trait-based models of phytoplankton ecology use the functional traits of individual species or entire communities to understand the biogeography, seasonal dynamics, and future responses of phytoplankton to environmental change [42,43]. Across taxonomic groups, major ocean regions, and marine and freshwaters, nearly all traits of ecological importance scale allometrically with cell size [38,44,45]. Despite its broad predictive power, however, theoretical and empirical studies have revealed complex interactions between cell size and environmental gradients such as temperature and nutrient supply, two of the principal abiotic factors structuring phytoplankton communities [16,46]. In general, large cells tend to dominate in the cold, nutrient-rich waters of high latitudes, and smaller cells are more abundant in lower latitudes, where temperatures are warmer and nutrient supplies are lower [16,47]. Other factors, such as grazing pressure, can interact with temperature and nutrients to modify size–abundance relationships [46]. Amidst a sea of trait correlations, the extent and complexity of these interactions make it difficult to infer causal relationships and develop a simple ecological model of abundance [39,41,42].

Across the tree of life, cell size is also correlated with the size of both the genome and the cell nucleus [24,34,48]. This relationship is commonly assumed to reflect simple packaging constraints, suggesting that over evolutionary timescales nucleus and cell sizes ebb and flow nonadaptively in response to changes in genome size [22,24,34]. Although intuitive, the mechanisms by which these 3 size components of the cell exert their influence on one another is unclear [22]. Alternatively, the strong associations between cell size and fitness-related traits, such as nutrient acquisition and growth rate, suggest cell size is an adaptive trait [23,24,34]. If larger cells require larger nuclei to balance space requirements for RNA synthesis in the nucleus and protein synthesis in the cytoplasm, then changes in the amount of bulk DNA are a means of modulating the size of the nucleus to maintain an optimal nuclear:cytoplasmic ratio (the "karyotopic ratio") [23]. Our novel approach to these questions—combining phylogenomics, empirical growth rates, and a global DNA metabarcoding database—highlighted a central role for genome size in the cellular and ecological properties of marine diatoms.

Although previous flow cytometry studies found correlations between genome size and cell size in diatoms [49,50], the genome sequences analyzed here identified repetitive DNA as the principal driver of genome size evolution. Nucleotypic effects describe the phenotypic changes that occur in response to changes in genome size [51]. In the diatoms studied here, repeat-driven changes in genome size over the past 100 million years had strong nucleotypic effects on 2 fitness-related traits—cell size and maximum growth rate (Figs 1 and 2). The same nucleotypic effects operate on microevolutionary timescales in diatoms as well. A comparison of 2 populations of the marine planktonic diatom, *Ditylum brightwellii*, with 2-fold difference in genome size and 4-fold difference in maximum cell volume, showed that the population with a smaller genome and cell size had a higher growth rate, and that genome size had a significantly greater (negative) impact on growth than cell size [52]. Larger genomes take longer to replicate, lengthening mitosis and cell doubling time [22,51]. Larger genomes also require additional investments of N and P to replicate and maintain, so in species with large genomes, these 2 essential nutrients cannot be allocated to RNA, ribosomes, and proteins, reducing growth rates and, over longer timescales, selecting for smaller genomes under conditions of nutrient limitation [41].

Although cell size is often considered a "master" phytoplankton trait, our results highlight the ecological importance of genome size as well. Genome size was not driven by increases in the amount of functional DNA, either through gene or genome duplications, but instead through changes in the amount of nonfunctional sequences, highlighting bulk DNA content as a phenotype with far-reaching consequences for phytoplankton physiology and ecology. Although genome size could be interpreted as an adaptive trait in this context [24], this must

be weighed against the deleterious effects of excess DNA, including mobile elements that can disrupt functional genes [53] and the metabolic burden of noncoding DNA [41]. Although evidence for this hypothesis is mixed, [54–56], the inclusion of population genetic parameters in our models might have shown whether diatoms with smaller effective population sizes are potentially more susceptible to nonadaptive genome expansions due to genetic drift [53]. Whatever the cause, our results provide support for a simple model in which many ecologically important traits, though perhaps more proximally related to cell size, are perhaps ultimately attributable to the size of the genome.

Ecologists have identified numerous biotic and abiotic factors that explain organismal abundance and geographic distributions [7–11]. Indeed, diatom abundance is shaped by abiotic factors such as temperature, along with both bottom-up effects such as nutrient supply, and top-down effects such as grazing pressure [57–59]. The data presented here showed that diatoms with larger genomes and, by extension, larger cell volumes are more abundant in regions and latitudes that experience colder temperatures, supporting Bergmann's rule and reinforcing a broader biogeographic trend of larger phytoplankton in colder seas [42]. This pattern has been attributed to temperature and numerous covarying factors [18]. For example, larger genomes and cells require more nutrients, which are generally in greater supply at higher latitudes [16,47]. In addition, grazing marine copepods have larger body sizes in colder temperatures [60], which might select for increased genome and cell size in colder parts of the ocean. Although including these and other factors in our models undoubtedly would have explained more of the variation in abundances, genome size predicted abundance remarkably well. Like most studies, the strong effect of latitude on the association between genome size and abundance reflects the context-specific nature of this association, which is typical of many ecological patterns [61]. Finally, a field study of freshwater benthic diatoms from geothermally heated streams found no evidence for Bergmann's rule [62], suggesting possible differences in diatom size–abundance relationships across ecosystem types or phylogenetic lineages.

Documenting abundance associations at a global scale is not without challenges. For example, in many cases the *Tara* Oceans samples represented a single snapshot in time of abundance at a location, precluding estimates of sampling error and potentially missing rapid seasonal changes in species abundance. Although these types of temporal limitations are common in spatial datasets that are global in scale, they have nevertheless proven to be extremely powerful in revealing broad ecological trends [40,63]. The diatom lineage studied here, Thalassiosirales, was well represented throughout the *Tara* Oceans samples and allowed us to uncover strong evidence linking genome size, temperature, maximum population growth rate, and species abundance [32,33]. Our results are consistent with size–abundance relationships found in large-scale phytoplankton studies [16,46], which might also be driven ultimately by genome size.

Similar associations have been found in multicellular organisms as well, suggesting genome size may shape patterns of species abundance broadly across the tree of life. Genome size has been linked to the distribution of flowering plants along a temperature gradient in the British Isles [64] and was positively correlated with regional abundance in 436 herbaceous plant species across Europe [28]. Although not related to a temperature gradient, salamanders are among the most abundant animal groups in many terrestrial ecosystems [65], and they have among the largest known genomes in the vertebrate lineage [66]. Like diatoms, much of the variation in genome size in these and other groups is attributable to noncoding sequences. Notably, despite the inherent difficulty in estimating abundance at a global scale, the amount of variation in abundance explained by genome size in our study was substantially greater than the typical range of variation accounted for in many ecological studies [67], highlighting the seemingly outsized role of genome size in the ecology of unicellular organisms.

In addition to the ecological consequences, our results highlight the unique cellular trade-offs imposed by changes in genome size in diatoms. Diatoms reproduce asexually throughout most of their life history and are unusual in that one of the 2 daughter cells following a mitotic event is smaller than the parent, leading to a reduction in the average diameter of a cell lineage over time, eventually triggering sexual reproduction and restoring the maximum cell size [68]. Although not measured here, nucleus size is positively correlated with genome size across the tree of life [24,69,70], and the same correlation likely exists for diatoms. With a fixed genome size that constrains the size of the nucleus, diatoms must optimize their surface area:volume ratio as cell size decreases across generations. Diatoms have vacuoles that function in buoyancy control, nutrient storage, and optimization of the surface area:volume ratio. Vacuoles occupy as much as 90% of the cell volume [57], and vacuole size can be modulated in response to environmental conditions [71]. The strong correlation between vacuole size and cell volume has led to the hypothesis that the vacuole has played a key adaptive role in diatom evolution by facilitating increases in cell size as a way to escape grazing pressure [57,58]. This hypothesis does not account for the parallel influence of genome size on cell size confirmed here.

Just as the discovery here of a directional effect of genome size on cell size highlights its lack of consideration from previous models, it likewise highlights the absence of several traits from our study. Although suggested by our models, increases in genome size may not be the proximal cause of increased cell volume. Genome size might affect cell volume indirectly, via upward pressure on nuclear volume or another latent character. In addition, although the functions of the vacuole as they relate to cell size are clear [58], vacuole size is a more labile trait, and it is unclear whether the vacuole exerts a causal influence on cell size or vice versa. The genome, nucleus, and vacuole have different functional roles in relation to cell volume, and all incur costs to maintain [57,72], so with a fixed genome size that presumably constrains the minimum size of both the nucleus and the cell, diatoms probably rely primarily on adjustments to the size and contents the vacuole as the ecological setting (e.g., rates of nutrient uptake, sinking rate, susceptibility to grazers) changes in response to decreases in the volume of a cell lineage over time.

Overall, the results presented here advance our understanding of species abundance by showing that a single emergent trait fundamental to all life, the size of the genome, can predict population abundance at a global scale. Moreover, the geographic variation in this pattern is entirely consistent with longstanding ideas regarding size–abundance associations in relation to the thermal environment. The addition of ecological information and other trait data to genome size estimates would likely generate a more informative model of species abundance, and this remains an important next step. Integrative approaches such as the one developed here, combining the seemingly disparate subdisciplines of phylogenomics and population ecology, may prove useful in forecasting widespread changes in the abundance of diatoms in response to ongoing climate change, especially in polar regions.

## Materials and methods

### Strain collection and culturing

Strains were collected from a variety of locales in the United States and isolated into monoclonal cultures. Additional strains were acquired from the National Center for Marine Algae and Microbiota (NCMA) in the United States or the Roscoff Culture Collection (RCC) in France. Marine strains were grown in L1 medium [73] and freshwater strains were grown in WC medium [74]. Cells were grown in batch culture at varying temperatures from 5 to 21°C on a 12–12 hour light–dark cycle. Cultures newly isolated in the Alverson Lab have been submitted

to the public culture collections at NCMA and The University of Texas Culture Collection of Algae (UTEX) (S1 Table).

## Draft genome sequencing, assembly, and gene prediction

See Supplementary File S1 of [35] for a detailed description of DNA extraction, sequencing, draft genome assembly, and gene model prediction. We additionally downloaded short read files for additional Thalassiosirales and Lithodesmiales (outgroups) strains that were available from the NCBI Sequence Read Archive (S1 Table) [75–78]. For these reads, we used a similar workflow as outlined in [35]. Briefly, we trimmed the reads using *Trimmomatic* v.0.36 [79], corrected the trimmed reads with *BayesHammer* [80], assembled the corrected reads with *SPAdes* v.3.12.0 [81], and removed contaminant contigs using *Blobtools* v.1.1.1 [82]. We removed contigs that had taxonomic assignment to bacteria, archaea, or viruses, or were shorter than 1 kb.

We also downloaded the reference genomes for *Chaetoceros tenuissimus* v.1 [83], *Cyclotella cryptica* v.2 [84], *Cyclotella nana* v.4 [85] (DOI: 10.5683/SP2/ZDZQFE), *Skeletonema marinoi* RCC75 v.1 [86], *Skeletonema marinoi* RO5AC v.1.1.2 [87] (DOI: 10.5281/zenodo.7786015), and *Thalassiosira oceanica* v.2 [88] (DOI: 10.5281/zenodo.4589594). When available, we also downloaded the predicted gene models and associated short reads from NCBI that were used in the genome assembly (S1 Table). For every genome with predicted gene models, we also calculated the average lengths of the genes, exons, and introns (S1 Table).

## Phylogenetic tree estimation

We used *BUSCO* v.5.1.3 [89,90] to estimate the completeness of each draft assembly using the stramenopiles_odb10 orthologs ($n$ = 100) (S1 Table). We used the detected single-copy orthologs to estimate a phylogenetic tree for all strains included in this study. First, we aligned the amino acid sequences of each ortholog using *MAFFT* v.7.505 [91] and the L-INS-i algorithm ("—localpair—maxiterate 1000"). Next, we trimmed the resulting alignments using *ClipKIT* v.1.3.0 [92] with the default smart-gap mode ("-m smart-gap"). We then concatenated all trimmed alignments into a supermatrix using the *pxcat* command in *Phyx* [93]. Finally, we estimated a phylogenetic tree from the supermatrix using *IQ-Tree* v.2.2.0.3 [94], partitioning the alignment by gene, using the LG+G substitution model, constraining the backbone topology to match the reference tree from [35], and calculating branch support with 10,000 ultrafast bootstrap replicates [95].

To generate an ultrametric, time-calibrated phylogeny, we estimated divergence times using *MCMCtree* in *PAML* v.4.9e [96]. We used nucleotide data from 4 loci: the nuclear *18S* and *28S*, and the plastid *rbcL* and *psbC*. We used the approximate likelihood approach [97], the GTR+G5 substitution model, and the independent rates clock model. Priors in the analysis were kept at their defaults. To ensure convergence in age estimates, we ran 2 independent MCMC chains of 2.5e6 generations, sampling every 200, with the first 5e5 discarded as burn-in. We checked that the effective sample size (ESS) of each parameter estimate was above 200 using *Tracer* v.1.7.2 [98].

We applied 8 calibrations to the reference tree in *MCMCtree* based on previous age estimates or fossil evidence. First, we placed upper and lower bounded constraints on the root (lower: 136 Ma; upper: 156 Ma) and the crown of Thalassiosirales–Lithodesmiales (lower: 115 Ma; upper: 141 Ma) based on estimates in [99]. We then placed a skewNormal prior distribution on the crown age of Thalassiosirales with a minimum age of 75 Ma based on the *Thalassiosiropsis* fossil [100,101]. Based on first fossil appearances in the Neptune marine micropaleontology database [102], we also placed lower bounds for the stem ages of

*Lithodesmium undulatum* (29.96 Ma), *Porosira glacialis* (9 Ma), and *Bacterosira constricta* (8.35 Ma). For *Cyclostephanos*, we placed a lower bound of 5 Ma on the crown age based on the fossil species *Cyclostephanos undatus* [101,103]. Lastly, we placed a lower bound of 6.15 Ma on the crown age of *Shionodiscus* based on the fossil species *Shionodiscus praeoestrupii* [101,104].

## Genome size estimation

We estimated haploid genome sizes using contaminant-free paired-end Illumina reads that aligned to the filtered draft genome assemblies. Briefly, we aligned the reads to the cleaned assembly using *minimap2* v.2.10 [105] using the presets for short reads and outputting the alignments in BAM format ("-ax sr"). We then extracted and kept only the aligned and correctly paired reads using the tool *bam2fastq* (https://github.com/jts/bam2fastq) with options "—aligned—no-unaligned—no-filtered." We estimated the genome size of each strain using 2 bioinformatic approaches based on *k*-mer [106] and read coverage histograms [107] (S4 Table).

For the *k*-mer-based genome size estimates, we used the script *kmercountexact.sh* from BBtools (https://sourceforge.net/projects/bbmap/). This script counts the number of unique *k*-mers in each pair of read files, estimates the genome size, and outputs a *k*-mer frequency histogram. We specified a range of *k*-mer lengths (17, 19, 21, 23, 25, 27, 29, and 31) because genome size estimates are highly dependent upon the chosen *k*-mer. For example, longer *k*-mers will collapse fewer short repeats and genome size estimates will be larger. We averaged the BBtools estimates for each *k*-mer length to produce a final genome size estimate for each strain. Because *kmercountexact.sh* can sometimes incorrectly identify the locations of the heterozygous and homozygous peaks in the histogram, we verified the accuracy manually in R [108]. We plotted the *k*-mer histograms in R to identify the peaks and calculate the estimated haploid genome size (*GS*) using the formula: $GS = N / (C * p)$, where *N* is the total number of genomic *k*-mers, *C* is the *k*-mer coverage, and *p* is the ploidy [106].

For read coverage-based estimates, we aligned the filtered read files to the draft assembly using *minimap2* and exported the alignments in BAM format. We then used the script *pileup.sh* from BBtools to calculate the total number of mapped base pairs and export a table of per-contig average coverage estimates. Next, we plotted the distribution of per-contig coverages in R and identified the mode of the distribution using the *asselin* function from the R package *modeest* [109]. To estimate the genome size, we used the Lander–Waterman formula [110]: $GS = LN / C$, where *L* is the read length, *N* is the total number of mapped reads, and *C* is the mode of the coverage distribution.

For *Lauderia annulata*, *Roundia cardiophora*, and *Thalassiosira punctigera*, the draft genome assemblies were too fragmented and incomplete to use for genome size estimates using the above approaches. Instead, we used the transcriptomes from these strains to estimate genome size in an approach adapted from [111]. Our approach to transcriptome assembly and contig filtering is detailed in Supplementary File S1 of [35]. In this approach, the filtered read files were aligned against the transcriptome assembly using *minimap2*, the resulting BAM file was parsed using *pileup.sh*, and the mode of the coverage distribution was estimated in R. The same formula for the read coverage-based approach is then used to estimate genome size.

## Ploidy estimation

We used *Smudgeplot* v.0.2.5 [112] to estimate the ploidy of each strain (S1 Table). This method uses short reads to disentangle genome structure and estimate ploidy. For this method, we used the sets of aligned and contaminant-filtered reads that were used in genome size

estimation. We used *jellyfish* v.2.3.0 [113] to count *k*-mers of length 21 from both forward and reverse reads. *Smudgeplot* then extracted the heterozygous *k*-mer pairs and calculated their coverages in order to estimate the ploidy of the sample.

### Repeat content estimation

We estimated the percentage of each genome that is composed of repetitive elements using the pipeline *DNApipeTE* v.1.3 [114] (S1 Table). This method allows for the fast assembly, quantification, and annotation of repeat sequences from a low-coverage sampling of reads. For each strain, we provided *DNApipeTE* with a single file of combined forward and reverse reads, the estimated genome size (in bp), and a custom repeat library for repeat annotation. The read files consist of the aligned and contaminant-filtered reads extracted previously for genome size estimation. We generated the custom repeat libraries for the genome assemblies using *RepeatModeler2* v.2.0.1 [115]. *DNApipeTE* performs sampling of the reads to produce low-coverage datasets to use during analyses. After initial testing of the pipeline using different coverages (0.01×, 0.05×, 0.1×, 0.25×), we determined that a coverage of 0.25× would be used. For some lower quality genomes with fewer available reads, a lower coverage (0.1×) was used. We estimated the repeat content for each strain using both the *k*-mer- and read coverage-based genome size estimations.

### Growth measurements

We calculated the maximum growth rates of strains using the relative chlorophyll *a* fluorescence (RF) measured on a Trilogy Fluorometer (Turner Designs, California, United States of America). In triplicate, we grew each strain at their maintenance temperature (5˚, 15˚, or 21˚ C) and measured their RF daily or every other day. We input the RF values into the R package *growthrates* [116] to calculate the maximum growth rate (μ) during exponential growth using the linear method [117]. We calculated the doubling time (in hours) of each strain by dividing μ by the natural logarithm of 2. We averaged the triplicate doubling times to get a strain average (S5 Table).

### Cell volume measurements

We collected information about minimum and maximum observed cell diameters and heights from biovolume databases and primary literature (S2 Table). For many marine species, we compiled cell volume data from the Helsinki Commission Phytoplankton Expert Group (HELCOM PEG) dataset (https://helcom.fi/helcom-at-work/projects/peg/) or [118]. For the remaining species, cell volume was calculated using size ranges reported in the primary literature. The height of cells is rarely reported, making it difficult to calculate cell volume using real measurements. We therefore used an approach to calculate cell heights using a ratio, *height = diameter * 0.5* [118]. For *Skeletonema*, we made a simplifying assumption that cell height equals cell diameter [119,120]. We then calculated the minimum and maximum cell volumes using appropriate equations for the approximate geometric shape of each species (S2 Table).

### Taxonomic assignment of OTUs

We downloaded the assembled and clustered V9-*18S* rDNA OTU sequences and read count abundances from the *Tara* Oceans expedition from Zenodo (DOI: 10.5281/zenodo.3768510) [121,122]. We initially filtered the OTUs to only those that had previous taxonomic assignments to the Thalassiosirales and Lithodesmiales. We aligned these filtered OTU sequences to

a set of reference *18S* rDNA sequences using *ssu-align* (http://eddylab.org/software/ssu-align/). We estimated the *18S* reference tree using *IQ-Tree* with the TIM2+F+I+G4 model. We then used the evolutionary placement algorithm (EPA) [123] in *RAxML* v.8.2.11 [124] to place the OTU sequences onto the reference phylogenetic tree of our *18S* sequences.

After OTU placement, we used *Gappa* v.0.8.0 [125] to parse the resulting EPA jplace file and assign a taxonomy to each OTU using the computed likelihood weights. We used 2 approaches to assign taxonomy. First, we used "*gappa examine assign*" to compute a majority taxonomy for each internal node based on a consensus of its descendants. For example, if the consensus threshold is set to 0.5 and 4 descendants are labeled "A;B;C" and 3 are labeled "A;B; D," the inner label will get labeled "A;B;C." We chose a consensus threshold of 0.7 for this first approach. This first approach resulted in OTUs for 22 species that overlapped with our genome size dataset. Second, we used "*gappa edit accumulate*" which adds together the likelihood weight ratio of each placement downward toward the root until the accumulated mass at the basal branch reaches the defined threshold. This approach is useful to assess placements distributed across nearby branches of the reference tree when the reference contains multiple representatives of the same species. We chose a threshold of 0.5 for this second approach. The second approach resulted in OTUs for 28 species that overlapped with our genome size dataset.

## OTU abundances and associated metadata

Following the taxonomic assignments of the OTUs, we filtered the original OTU abundance count table down to only those assigned to the species in our dataset. To account for differences in read depth between the *Tara* Oceans samples, we transformed the OTU counts to proportions prior to statistical analyses [126]. We used the R package *dplyr* [127] to aggregate the abundances for each species and calculate the sum of each species at each *Tara* sampling location. This generated a table with a single relative abundance measurement per species at each sampling locale. We then downloaded 2 tables containing the associated environmental metadata of each sample taken in the *Tara* Oceans expeditions from PANGAEA (DOI: 10. 1594/PANGAEA.858201, DOI: 10.1594/PANGAEA.875576) [128]. We combined the 2 metadata tables and merged the combined data with the abundances. This produced a final table containing abundances of each species and the metadata for each sampling locale.

## Statistics

The following variables were $\log_{10}$-transformed before statistical analyses: genome size, average exon length, average intron length, doubling times, and cell volumes. We applied arcsin transformation to the percent repeat content. No transformation was applied to average gene length, GC percent, temperature, and latitude.

We calculated the phylogenetic signal of genome size with Pagel's lambda [129] with the *phylosig* function in the R package *phytools* [130]. Ancestral state reconstruction of minimum cell volume was estimated using maximum likelihood under an Ornstein–Uhlenbeck model with the *anc.ML* function in the R package *phytools*. We calculated Spearman's rho between the *k*-mer- and read coverage-based genome size estimates using the R stats function *cor.test*. To test for the correlation between variables, we performed linear regression using the R stats function *lm* and PGLS using the functions *comparative.data* and *pgls* from the R package *caper* [131]. The phylogenetic tree used in PGLS was the ultrametric tree estimated above.

We performed phylogenetic path analysis [132] using the function *phylo_path* in the R package *phylopath* [133]. Fourteen models were designed and tested to determine if genome size had no effect (null models), direct effects (direct models), or indirect effects (indirect

models) on doubling time (S7 Fig). We used the default evolutionary model of "lambda" and calculated averaged models using the "full" approach [132].

We tested 3 hypotheses that species abundances are predicted by the interaction of genome size and ocean region, temperature, or latitude. Prior to model fitting, we centered all continuous variables in these models, except latitude and relative abundance. We fit these models using Bayesian phylogenetic multilevel models, estimating the posterior distributions of each model using *rstan* [134] via the *brms* package [135]. Species abundances (the response variable) were modeled using a gamma distribution with a log link function. Prior sensitivity was assessed by running models with uninformative or weakly informative priors. To account for phylogenetic non-independence between species in our models, we calculated a covariance matrix from the ultrametric phylogenetic tree using the function *vcv.phylo* in the R package *ape* [136]. We calculated the mean and 95% credible intervals for each model parameter and each derived quantity, from the joint posterior distribution of the models. Model effects were considered significant if the means and 95% credible intervals did not overlap zero [137]. Predictions for the interactive effect of latitude, temperatures, or ocean region with genome size on abundances were made from the models using the *predictions* function in the R package *marginaleffects* [138]. For latitude effects, we fit generalized additive models to account for nonlinear effects of latitude on species abundance. Each model in *brms* was run using 2 Markov chains for 10,000 iterations. We assessed chain convergence using potential scale reduction factors [139] and model fit using Bayesian $r^2$ [140]. In addition, we assessed model fits using posterior predictive checking.

We plotted figures, phylogenetic trees, and maps using the R packages *ggplot2* [141], *aplot* [142], *ggtree* [143], *maps* [144], and *ggmap* [145]. Figure edits were made using Adobe Illustrator.

## Supporting information

**S1 Fig. Strong correlation between haploid genome size estimates.** Scatterplot showing the correlation between estimated haploid genome sizes via *k*-mer- and coverage-based approaches. The black line indicates the regression coefficient. Spearman's rho and associated *P* value are shown above the plot. The data and code to generate this figure can be found in https://doi.org/10.5281/zenodo.12608914.
(TIF)

**S2 Fig. No difference in genome size between marine and freshwater diatoms.** Violin plots showing the distribution of coverage-based haploid genome size estimates for marine (black) and freshwater (blue) species. The test statistic and *P* value from a Wilcoxon rank sum test are shown above the plot. The data and code to generate this figure can be found in https://doi.org/10.5281/zenodo.12608914.
(TIF)

**S3 Fig. Genome size is predicted by repetitive elements, gene lengths, and GC content.** Phylogenetic generalized least squares (PGLS) models predicting genome size by **(A)** the estimated percentage of repetitive elements in the genome, **(B)** the average gene length, **(C)** the average exon length, **(D)** the average intron length, and **(E)** the genome average GC content. Black lines show the estimated regression coefficient. The PGLS $r^2$ and associated *P* value are shown above each plot. The data and code to generate this figure can be found in https://doi.org/10.5281/zenodo.12608914.
(TIF)

**S4 Fig. Percentages of different repetitive element classes in each genome.** **(A)** A time-calibrated phylogeny of the diatom order Thalassiosirales, modified from [35]. Strain numbers follow the species names. **(B)** Stacked bar plot showing the percentage of each genome belonging to different repetitive element classes as estimated using *dnaPipeTE*. Colors denote the different repeat classes and gray represents the percentage of the genome that is non-repetitive. Abbreviations: DNA, DNA transposons; LINE, long interspersed nuclear elements; LTR, long terminal retroelements; RC, rolling circles or *Helitrons*. The data and code to generate this figure can be found in https://doi.org/10.5281/zenodo.12608914.
(TIF)

**S5 Fig. Genome size is predicted by minimum and maximum cell volumes.** Phylogenetic generalized least squares (PGLS) models predicting genome size by the **(A)** minimum and **(B)** maximum calculated cell volume. Black lines show the estimated regression coefficient. The PGLS $r^2$ and associated $P$ value are shown above each plot. The range of cell volume sizes (minimum to maximum) are shown for each species on **(C)** linear and **(D)** $\log_{10}$-transformed scales. The data and code to generate this figure can be found in https://doi.org/10.5281/zenodo.12608914.
(TIF)

**S6 Fig. Doubling time is predicted by genome size and temperature.** Phylogenetic generalized least squares (PGLS) model predicting doubling time by the additive effects of genome size and temperature. The black line shows the estimated regression coefficient. The points are colored according to the growth temperature of the strain. The PGLS $r^2$ and associated $P$ value are shown above each plot. The data and code to generate this figure can be found in https://doi.org/10.5281/zenodo.12608914.
(TIF)

**S7 Fig. Models tested in the phylogenetic path analyses.** Directed acyclic graphs showing the 14 models tested in the phylogenetic path analysis. Models were defined to test if genome size had no effect (null models), direct effects (direct models), or indirect effects (indirect models) on doubling time. Temperature, GC percentage, and minimum cell volume are also included as variables in the models. Arrow direction indicates the causal relationship being tested between 2 variables. The data and code to generate this figure can be found in https://doi.org/10.5281/zenodo.12608914.
(TIF)

**S8 Fig. The best and average models from the phylogenetic path analyses.** Results of the phylogenetic path analysis. **(A, B)** Best models and **(C, D)** averaged models using coverage-**(A, C)** or *k*-mer-based **(B, D)** genome size estimates. Arrow color and width represent the direction and magnitude of regression coefficients, indicated by numeric labels (positive: blue; negative: red; nonsignificant: gray). Full lines show coefficients that differ significantly from 0, whereas negative lines overlap with 0 and are nonsignificant. The data and code to generate this figure can be found in https://doi.org/10.5281/zenodo.12608914.
(TIF)

**S9 Fig. Latitude, ocean region, and temperature interact with genome size—abundance relationships.** Results from alternative taxonomic assignment of barcodes for 22 diatom species across 210 sampling stations from the *Tara* Oceans expedition. Panels show Bayesian multilevel regression models predicting relative abundance by the interaction of genome size with **(A)** latitude, **(B)** ocean region, or **(C)** temperature. Nonlinear effects of latitude were modeled in **(A)** using a generalized additive model. Significant estimates for the Arctic, Southern, South

Pacific, and North Pacific Oceans are shown with solid lines in **(B)**. Nonsignificant estimates for the other ocean regions are shown with dotted lines in **(B)**. Although all predictors are treated as continuous in **(C)**, we used the model to predict the interactive effect of 4 discrete temperatures (0, 10, 20, and 30˚C) with genome size on relative species abundance. The Bayesian $r^2$ for each model is indicated above each panel. The data and code to generate this figure can be found in https://doi.org/10.5281/zenodo.12608914.
(TIF)

**S1 Table. Summary of strain collection, accession numbers, and genome characterization.**
(XLSX)

**S2 Table. Summary of the calculated minimum and maximum cell volumes.**
(XLSX)

**S3 Table. Results of the phylogenetic path analyses.**
(XLSX)

**S4 Table. Results of genome size estimation.**
(XLSX)

**S5 Table. Results of growth experiments and doubling time calculation.**
(XLSX)

## Acknowledgments

We thank Jeremy Beaulieu for comments on an earlier version of the manuscript. Simon Tye helped design and implement Fig 1. The used resources were available through the Arkansas High Performance Computing Center, which is funded through multiple NSF grants and the Arkansas Economic Development Commission.

## Author Contributions

**Conceptualization:** Wade R. Roberts, Adam M. Siepielski, Andrew J. Alverson.

**Data curation:** Wade R. Roberts.

**Formal analysis:** Wade R. Roberts, Andrew J. Alverson.

**Funding acquisition:** Andrew J. Alverson.

**Investigation:** Wade R. Roberts, Adam M. Siepielski, Andrew J. Alverson.

**Methodology:** Wade R. Roberts, Adam M. Siepielski.

**Project administration:** Wade R. Roberts, Andrew J. Alverson.

**Resources:** Andrew J. Alverson.

**Supervision:** Andrew J. Alverson.

**Visualization:** Wade R. Roberts.

**Writing – original draft:** Wade R. Roberts, Andrew J. Alverson.

**Writing – review & editing:** Wade R. Roberts, Adam M. Siepielski, Andrew J. Alverson.

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
