## [Editor Report · Decision Letter 0]

26 Feb 2024

Dear Dr Roberts, 

Thank you for submitting your manuscript entitled "Genome size predicts diatom abundance in the ocean" for consideration as a Research Article by PLOS Biology.

Your manuscript has now been evaluated by the PLOS Biology editorial staff, as well as by an academic editor with relevant expertise, and I'm writing to let you know that we would like to send your submission out for external peer review.

IMPORTANT: We would like to review your paper as a Short Report. Because your manuscript is already concise, there's no need for any re-formatting, but please select "Short Reports" as the article type when you upload the additional metadata (see next para).

Once your full submission is complete, your paper will undergo a series of checks in preparation for peer review. After your manuscript has passed the checks it will be sent out for review. To provide the metadata for your submission, please Login to Editorial Manager (https://www.editorialmanager.com/pbiology) within two working days, i.e. by Feb 28 2024 11:59PM.

Kind regards,

Roli Roberts

Roland Roberts, PhD

Senior Editor

PLOS Biology

rroberts@plos.org

---

## [Decision Letter · Decision Letter 1]

16 Apr 2024

Dear Dr Roberts,

Thank you for your patience while your manuscript "Genome size predicts diatom abundance in the ocean" was peer-reviewed at PLOS Biology. It has now been evaluated by the PLOS Biology editors, an Academic Editor with relevant expertise, and by three independent reviewers. 

You'll see that reviewer #1 is broadly quite positive, but wants you to improve your coverage of the prior literature, and tighten up your thinking around causality throughout (s/he returns to this theme several times re the causal relationship between genome size, nucleus size and cell size). Reviewer #2 finds your Title misleading, asks for some stats support, to clarify your treatment of ploidy, and asks if you could use a single copy gene for the Tara analysis. Reviewer #3 is very positive, but s/he also questions the direction of causality (wanting you to discuss possible selection mechanisms for your proposed direction), and wants you to include nutrient concentration as a variable (if data are available).

I discussed these comments with the Academic Editor, who agreed that we should invite you to revise the work to thoroughly address the reviewers' reports.

Given the extent of revision needed, we cannot make a decision about publication until we have seen the revised manuscript and your response to the reviewers' comments. Your revised manuscript is likely to be sent for further evaluation by all or a subset of the reviewers.

**IMPORTANT - SUBMITTING YOUR REVISION**

*Re-submission Checklist*

*Published Peer Review*

*PLOS Data Policy*

*Blot and Gel Data Policy*

Sincerely,

Roli Roberts

Roland Roberts, PhD

Senior Editor

PLOS Biology

rroberts@plos.org

REVIEWERS' COMMENTS:

Reviewer #1:

[identifies himself as Adam B. Roddy]

This manuscript assesses the relationships between genome size, genome structure, cell size, population growth rate, relative abundance among a group of diatoms by combining laboratory experiments, field surveys, sequencing, and phylogenetic and path analyses. I particularly like the breadth of these approaches that are presented succinctly in one paper. Indeed, it is rare to see a paper that tackles all of these issues at once, and I think that this paper did this very well on the whole.

The paper was also well-written and easy to follow (except see below for suggestions to setup the results and discussion better). However, I do discuss below a few issues that I found concerning.

First (and easily addressable) is that the manuscript cites very little previous literature (particularly in the introduction), including highly relevant literature. In one Google Scholar search, I found a paper about temperature effects on cell size in diatoms and its implications for Bergmann's rule, but this paper was not cited in this manuscript. I find it odd to have missed such basic information. Furthermore throughout the manuscript, from the very beginning of the introduction to the very end of the discussion, there are very broad statements made that lack citation. Stating a generally accepted principle is one thing, but saying something like "Theoretical studies have developed" or "Another equally large body of literature has sought" without providing a single citation to even a review paper is odd.

Second, I think the logic describing presumably causal relationships is confused and muddled throughout the manuscript. (I reference some of these below.) For example, having a larger cell does not mean that there must necessarily be a larger genome. Nor does having a larger genome necessarily that a mature cell will be larger. Rather, genome size determines the minimum cell size, and maximum mature cell size would determine the maximum possible genome size. Moving beyond thing about regression relationships (which show a line through the mean of the population) and thinking more about the outer limits of the point clouds would be useful. To be clear, I don't think that the way these ideas are discussed is any worse in this manuscript than in most other papers. And I am happy that this manuscript performs a path analysis, even if this analysis is based on mature cell size or mean traits. (Our statistics are fundamentally limited to address the effects of genome size on cell size because our methods predominantly are focused on elucidating mean effects and not the limits of data. This is why I argue below for having clear inductive reasoning that is independent of statistical results [which are inherently deductive].)

pg 3 para 1: These references to the literature could be better supported by, well, references to the literature. This is particularly notable given that only 5 papers are cited in the entire Introduction section.

Diatoms are outside my normal field, but I do find strange how little reference to prior work there is in this manuscript. For example, Adams et al. (2013 GCB, "Diatoms can be an important exception to temperature-size ruels at species and community levels of organization") is not cited, even though it uses a quasi-experimental approach to test whether temperature impacts diatom cell size, and, by extension, whether diatoms may follow Bergmann's Rule. (Notably, they do not sample across latitudes.) Thus, while this manuscript is easy-to-read and clearly written, I am concerned that important prior literature has been ignored.

pg 6, 3 lines up from bottom: is there an extra "top" in this sentence?

pg 7, 8 lines up: there is an extra "either" in this sentence

pg 8, bottom: Thsi description of the relationship between genome size and cell size is simplistic. First, while most studies (including the present one) typically present correlations between genome size and mature cell size, there is ample evidence suggesting there is a mechanistic relationship. Specifically, genome size (by extension, nuclear volume) determines the minimum possible cell size. However, mature cells can be larger than this minimum cell size. In plants, for example, genome size is an exceptionally strong predictor of meristematic cell volume, but only a weak predictor mature cell volume...precisely because mature cells can be larger than their minimum cell size. Thus, genome size does act as a hard, physical limit on cell size, but only on minimum cell size.

It might be possible to estimate the minimum cell size from genome size for these diatoms. How close mature cell sizes are siting to this minimum cell size bound likely reflects the how strongly selection on cell size would impact minimum cell size and thus genome size.

I am not an expert on diatoms, and my thinking is based on plants. But, even if there are no changes in genome size, cell size can change (because mature cells are larger than minimum cell size)-and this variation in mature cell size CAN be adaptive.

It's also worth noting that because the effects of genome size on nucleus size are extremely tightly linked physically, (on my opinion) the only relevant point about adaptation in this relationship is that nucleus size remains just a bit larger than the genome instead of growing much larger than the genome and being tied more to fluctuations in cell size.

It's vitally important when talking about genome size, nuclear volume, and cell size to be very specific and clear about the causal/mechanistic relationships. This is why correlations (such as those presented in Figure 1 and in most other papers) are merely correlations and do not necessarily describe any physical relationship between genome size and cell size. Even if phylogenetic regressions between these variables are statistically significant, that relationship still does not say anything about the physical relationship between the two traits; rather, it is about how the coevolve. And trait coevolution could, in theory, be driven largely by a physical mechanism or result from selection acting on both traits. Inductive reasoning-and not only statistical relationships-would be required to describe the causal relationships.

pg 9, line 3: When I read "laboratory growth experiments", I was at first confused. Admittedly, I had not yet read the methods since they are at the end. I suspect that the methods will be at the end in the final version of the paper. In that case, I think it would be worth either in the Introduction or in the Results giving a little shout-out or sign-posting so that the reader knows what data are being used. I had assumed that growth rate data were compiled from the literature, so clarifying it was measured would help.

pg 9, para 2 line 1: This sentence sets up a false dichotomy. By definition, what makes genomes bigger is having more nucleotides. A larger cell is not a mechanistic explanation for larger genomes. Rather, it is a correlation, or a result of having a larger genome, but a larger cell per se cannot tell us anything about the size of the genome EXCEPT the maximum size of the genome (i.e. the genome cannot be, by definition, larger than the entire cell).

pg 9, last para: I find the logic here confusing. It is a strange argument to say that genome size is more important than cell size to an organism's ecology. Does genome size itself limit rates of diffusion into the cell? Does genome size per se determine how many cells can be packed into a space? The answer to both questions is no. What CAN be said is that genome size is a stronger correlate of ecology than cell size. But that is a different point: in our subjective position of trying to characterize an organism's ecology, genome size is very useful...in contrast to saying that the diatom's way of life is more defined by the size of the genome than by the size of its body. 

pg 19 top: Evidence supporting Lynch's ideas about Ne and genome size is severely lacking. Multiple papers in other systems (fishes, mammals) have tested for a relationship between Ne and population size and find no relationship.

pg 11 middle: This paragraph implies a positive correlation between genome size and abundance in vertebrates (based only on salamanders). Doesn't that contradict the relationship reported in this paper (restricting to the black points from the lower latitudes where salamanders are also predominant)?

pg 11, last line: "nucleus size is positively correlated with genome size across the tree of life"...where is the citation to this statement?

I hope the authors have found these comments constructive and useful. I commend them for what is, on the whole, a very good paper.

Sincerely,

Adam B. Roddy

Reviewer #2:

This manuscript presents analyses of an impressive number of novel diatom assemblies (53).

The authors analyze 53 original genome assemblies' sequences and 14 previously published assemblies of 51 species of diatoms from the Thalassiosirales order. They use this dataset to investigate the relationship between genome size, different genomic traits, growth rates, and the relative abundance of diatoms in the ocean.

They report a positive linear relationship between genome size and abundance in the Arctic and Southern Oceans (Figure 3D), whereas all other regions either had no or a negative effect of genome size on species abundance. 

It seems very misleading to me that the title and abstract of this manuscript suggest that genome size predicts Thalassiosirales abundance in the ocean.

Major comments

1. Please estimate the significance of the relationships between genome size and relative abundance of species in Figure 3C. It is likely there is no 

2. P5. "there was no association between genome size … with the presence of polyploidy". The authors seem to confuse the actual genome size of a species (that would take ploidy level into account) and the haploid genome size. The latter can be estimated from an assembly when the two parenral chromosome copies are not too polymorphic. Please clarify.

3. P19. It is not clear to me how the relative abundance of each species was estimated from the Tara dataset. Usually, relat

---

## [Decision Letter · Decision Letter 2]

26 Jun 2024

Dear Dr Roberts,

Thank you for your patience while we considered your revised manuscript "Genome size predicts diatom abundance in the polar oceans" for publication as a Short Report at PLOS Biology. This revised version of your manuscript has been evaluated by the PLOS Biology editors, the Academic Editor and one of the original reviewers.

Based on this review and on our Academic Editor's assessment of your revision, we are likely to accept this manuscript for publication, provided you satisfactorily address the following data and other policy-related requests:

IMPORTANT - Please attend to the following:

a) Please flip the title round and change it to "Diatom abundance in the polar oceans depends on genome size" (if you think that "depends on" is too strong for the evidence, then maybe "Diatom abundance in the polar oceans is predicted by genome size").

b) Please address my Data Policy requests below; specifically, we need you to supply the numerical values underlying Figs 1ABCDE, 2B, 3ABCDE, S1, S2, S3ABCDE, S4AB, S5ABCD, S6, S9ABC, either as a supplementary data file or as a permanent DOI’d deposition. I note that you already have an associated Zenodo deposition (DOI:10.5281/zenodo.10610738), which looks very thorough. Please could you confirm that this deposition contains sufficient data and code to recreate the Figures?

c) Please cite the location of the data clearly in all relevant main and supplementary Figure legends, e.g. “The data and code needed to generate this Figure can be found in https://zenodo.org/records/10610738"

d) I note that you mention the reviewers ("We thank... three reviewers") in the Acknowledgements. While we appreciate the sentiment, this is against PLOS policy, so please could you remove this?

We expect to receive your revised manuscript within two weeks. 

*Published Peer Review History*

*Press*

Sincerely,

Roli Roberts

Roland Roberts, PhD

Senior Editor

rroberts@plos.org

PLOS Biology

DATA POLICY:

Regardless of the method selected, please ensure that you provide the individual numerical values that underlie the summary data displayed in the following figure panels as they are essential for readers to assess your analysis and to reproduce it: Figs 1ABCDE, 2B, 3ABCDE, S1, S2, S3ABCDE, S4AB, S5ABCD, S6, S9ABC. NOTE: the numerical data provided should include all replicates AND the way in which the plotted mean and errors were derived (it should not present only the mean/average values).

CODE POLICY

DATA NOT SHOWN?

REVIEWERS' COMMENTS:

Reviewer #1:

[identifies himself as Adam Roddy]

I previously reviewed an earlier submission of this manuscript. I am satisfied with the edits made since the first submission, and I appreciate the authors' patience with me as many of my concerns on the previous version were based on my own biases due to specializing in plants.

---

## [Editor Report · Decision Letter 3]

3 Jul 2024

Dear Dr Roberts,

Thank you for the submission of your revised Short Report "Diatom abundance in the polar oceans is predicted by genome size" for publication in PLOS Biology. On behalf of my colleagues and the Academic Editor, Andrew Tanentzap, I'm pleased to say that we can in principle accept your manuscript for publication, provided you address any remaining formatting and reporting issues. These will be detailed in an email you should receive within 2-3 business days from our colleagues in the journal operations team; no action is required from you until then. Please note that we will not be able to formally accept your manuscript and schedule it for publication until you have completed any requested changes.

Sincerely, 

Roli Roberts

Senior Editor

PLOS Biology

rroberts@plos.org